# Cost-effectiveness analysis protocol of the Smart Triage program: A point-of-care digital triage platform for pediatric sepsis in Eastern Uganda

Edmond C. K. Li[1], Sela Grays[2]*, Abner Tagoola[3], Clare Komugisha[4], Annette Mary Nabweteme[4], J. Mark Ansermino[2], Craig Mitton[1,5], Niranjan Kissoon[6], Asif R. Khowaja[7]

1 School of Population and Public Health, University of British Columbia, Vancouver, British Columbia, Canada, 2 Department of Anesthesiology, Pharmacology & Therapeutics, University of British Columbia, Vancouver, British Columbia, Canada, 3 Jinja Regional Referral Hospital, Jinja, Uganda, 4 Walimu, Coral Crescent, Kololo, Kampala, Uganda, 5 Centre for Clinical Epidemiology and Evaluation, Vancouver Coastal Health, Vancouver, Canada, 6 Department of Pediatrics, University of British Columbia, Vancouver, British Columbia, Canada, 7 Faculty of Applied Health Sciences, Brock University, St. Catharines, Canada

* sela.grays@bcchr.ca

**Funding:** This research was funded in whole, or in part, by the Wellcome Trust [Grant number 215695/B/19/Z] awarded to JMA. For the purpose

## Abstract

### Background

Sepsis is a clinical syndrome characterized by organ dysfunction due to presumed or proven infection. Severe cases can have case fatality ratio 25% or higher in low-middle income countries, but early diagnosis and timely treatment have a proven benefit. The Smart Triage program in Jinja Regional Referral Hospital in Uganda will provide expedited sepsis treatment in children through a data-driven electronic patient triage system. To complement the ongoing Smart Triage interventional trial, we propose methods for a concurrent cost-effectiveness analysis of the Smart Triage platform.

### Methods

We will use a decision-analytic model taking a societal perspective, combining government and out-of-pocket costs, as patients bear a sizeable portion of healthcare costs in Uganda due to the lack of universal health coverage. Previously published secondary data will be used to link healthcare utilization with costs and intermediate outcomes with mortality. We will model uncertainty via probabilistic sensitivity analysis and present findings at various willingness-to-pay thresholds using a cost-effectiveness acceptability curve.

### Discussion

Our proposed analysis represents a first step in evaluating the cost-effectiveness of an innovative digital triage platform designed to improve clinical outcomes in pediatric sepsis through expediting care in low-resource settings. Our use of a decision analytic model to link secondary costing data, incorporate post-discharge healthcare utilization, and model clinical

of open access, the author has applied a CC BY public copyright license to any Author Accepted Manuscript version arising from this submission. This study has been funded by the Wellcome Trust Innovator Award (https://wellcome.org/grant-funding). Study sponsors do not have and will not have a role in study design; collection, management, analysis, and interpretation of data; writing of the report; or the decision to submit the report for publication. They will not have ultimate authority over any of these activities.

**Competing interests:** The authors have declared that no competing interests exist.

endpoints is also novel in the pediatric sepsis triage literature for low-middle income countries. Our analysis, together with subsequent analyses modelling budget impact and scale up, will inform future modifications to the Smart Triage platform, as well as motivate scale-up to the district and national levels.

## Trial registration

*Trial registration of parent clinical trial*: NCT04304235, https://clinicaltrials.gov/ct2/show/NCT04304235. Registered 11 March 2020.

## Introduction

### Sepsis, the Importance of Triage, and the Smart Triage program

Sepsis is a syndrome characterized by an inflammatory state resulting from a presumed or proven infection that results in organ dysfunction and/or death [1], with global disease-specific mortality of 20% and a case fatality ratio of 25% or higher in low-middle income countries (LMICs) [2,3]. Earlier treatment results in better outcomes; therefore, international guidelines for timely treatment were shortened from 3 hours to 1 hour in 2016 [4]. Treatment comprises a "sepsis bundle" of antimicrobials, fluid, and oxygen as clinically indicated [5].

While high-income countries have employed evidence-based patient triage strategies to expedite sepsis treatment [6], these strategies involve various laboratory investigations rarely available in LMIC settings, where delays in treatment leading to a high case fatality ratio are commonplace [7]. LMIC-specific guidelines such as the Emergency Triage Assessment and Treatment (ETAT) [8] and others [9,10], show good predictive power for mortality and improve clinical outcomes when successfully implemented [11–13]. However, the complexity of these guidelines introduces challenges in everyday usage and staff training in an environment where patient burden and new-staff turnover is high [14,15]. Infrastructural barriers to patient care organization and medication shortages further delay sepsis treatment even when a sick child has been identified through effective triage [15].

The Smart Triage program [16] tackles guideline complexity via a prediction model for admission to facilitate the identification of children at risk of deterioration. This model classifies children into three levels of risk categories, using clinical variables found in ETAT and other triage studies that are then used as inputs in an algorithm derived from machine learning techniques. Our methods for initial consideration of variables have been published [17], and the details of the final algorithm are under peer review. This model was developed using data from the trial's pre-intervention phase, and has been integrated into a mobile application to reduce cognitive error and training time compared to a manual triage process. The triage results from the mobile application will be linked to a Clinician Dashboard local web page and a locally hosted Bluetooth treatment-tracking system to improve patient care organization. The Smart Triage program contains all of these components in a package, which will be implemented at the Jinja Regional Referral Hospital (JRRH) in Uganda, in an interventional trial for which the protocol is published [16].

### Economic evaluation of LMIC triage programs

When considering new programs in low-resource settings, economic evaluations such as cost-effectiveness analyses can inform policy decisions and motivate further research. While cost-

effectiveness analyses of quality improvement initiatives that deliver earlier sepsis care have demonstrated reduced cost and increased benefit over standard of care in high-income settings [18,19], such analyses are sparse for triage programs in LMICs. In the literature, studies may report partial costs without a formal economic evaluation [14], or perform cost-effectiveness analyses on intermediate outcomes without linkage to clinical endpoints [20]. One analysis used a robust decision-analytic model, but interventions evaluated spanned across the inpatient and post-discharge timeframes and were not isolated to the triage process [21]. Lastly, many analyses focused on the perspective of the healthcare system [13,20,22], but given the lack of universal health coverage in many LMICs, a societal perspective that also includes out-of-pocket (OOP) costs and time/productivity loses may be more appropriate. Therefore, in light of the literature gaps presented, we propose a cost-effectiveness analysis alongside the ongoing Smart Triage trial, using a decision analytic model, costing data from previous trials [23], linkage of intermediate outcomes to clinical endpoints [24], and a societal perspective. Our proposed economic evaluation serves to complement the previously published protocol of the Smart Triage trial [16], and will play a role in shaping policy and encouraging follow-up research surrounding the Smart Triage platform.

## Materials and methods

### Study objectives

1. Calculate the incremental cost relative to years-of-life lost (YLL) averted from the Smart Triage program compared to the standard of care, using the pre-intervention phase at JRRH as the control.

2. Determine the probability of cost-effectiveness of the Smart Triage program under different willingness-to-pay thresholds, when implemented in a government regional referral hospital in Uganda, using a decision-analytic model from the societal perspective. The comparator will be an ETAT-based manual system widely used in Uganda, including our study site prior to implementation of Smart Triage.

3. Investigate the impact of OOP costs on cost-effectiveness by secondary analyses from the government and patient perspectives.

### Description of parent clinical trial

The Smart Triage program and a corresponding clinical trial will be implemented in the pediatric outpatient department at JRRH in Uganda, which serves children up to age 18 presenting from the community at various degrees of disease severity. The outpatient department serves as a gateway to admission if necessary. All presenting children, except those with scheduled appointments, are eligible to participate. However, enrollment of all children is not possible due to a high patient load, and therefore a sampling framework as published in the parent trial's protocol is used to obtain a representative sample [17]. The Smart Triage trial was designed as a difference in differences trial, with a Kenyan site for control. However, we will restrict this cost-effectiveness analysis to the Ugandan site at JRRH, to minimize the impact on our analysis due to variations in baseline ETAT implementation between countries. Therefore, we will compare the post-intervention data at JRRH with that of the pre-intervention phase, with the latter serving as control for our analysis.

The primary outcome of the Smart Triage trial is the difference in the proportion of children who receive a sepsis bundle of care within one hour before and after Smart Triage's

implementation. Enrolled children will be followed up to assess in-hospital mortality (if admitted), 7-day post-discharge mortality, and 7-day post-discharge healthcare utilization rates. Of note, Smart Triage does not aim to indiscriminately increase the number or proportion of children receiving sepsis bundle. Indication for sepsis treatment is determined by local clinicians both before and after Smart Triage implementation. Smart Triage simply facilitates the timelier delivery of sepsis treatment, if deemed indicated by local clinicians, through a data-driven quality improvement process.

Further details of the parent clinical trial such as program implementation, clinical trial protocol, and sample size calculations have been published [16] with appropriate clinical trial registration on clinicaltrials.gov (NCT04304235). Ethics approval has been obtained from Makerere University School of Public Health (MUSPH) Higher Degrees, Research and Ethics Committee. MUSPH provided approval for the parent study and this economic evaluation to be conducted at Jinja Regional Referral Hospital in Jinja, Uganda. Written informed consent for the parent clinical trial as well as follow-up analyses such as this economic evaluation will be obtained from all caregivers/participants prior to enrollment. Patient recruitment has commenced and program implementation is underway, with the earliest projected end-date of data collection being late 2021. While the parent trial's protocol aims to establish clinical efficacy of the Smart Triage platform, this protocol aims to establish cost-effectiveness, and will be conducted alongside the parent clinical trial.

## Decision analytic model

We will perform the economic evaluation of Smart Triage using a decision analytic model (Fig 1). The model structure is based on a similar model used to evaluate the cost-effectiveness of a post-discharge program in Uganda named Smart Discharges [25], and the model care pathways have been shown to have face validity by collaborators at JRRH. As patients in Uganda bear a significant portion of healthcare costs due to the lack of universal healthcare, we will take a societal perspective for our base case analysis [26]. Also, since Smart Triage is a program that facilitates immediate treatment for an acute illness, it would likely exert its strongest effects on short-term outcomes, and therefore we will use a 7-day horizon for our analysis.

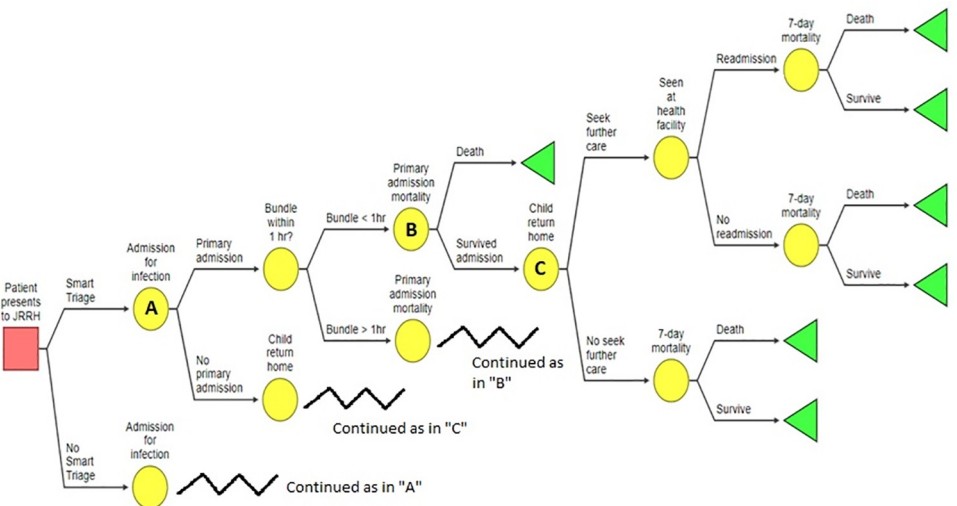

**Fig 1. Decision analytic model of post-triage and post-discharge care pathways, used for the cost-effectiveness analysis of Smart Triage.**

**Table 1. Key assumptions of the decision analytic model.**

| Assumption | Rationale |
|---|---|
| Smart Triage exerts its strongest effects on mortality and short-term morbidity within the initial 7 days. | Smart Triage facilitates an acute intervention to target short-term outcomes (sepsis-induced mortality and morbidity during the immediate illness) and will likely exert its strongest effects in this timeframe. The model may be updated when long-term disability data and disability weights for sepsis survivors become available in LMICs. |
| The odds ratio of death for receiving a sepsis bundle within 1 hour in high-income setting may be used to link the Smart Triage study's primary outcome to mortality in an LMIC setting. | LMIC-specific data in this regard is not yet available. However, due to various factors predisposing LMIC patients to a higher disease severity upon presentation and the lack of expensive intensive care support, a timely sepsis bundle may be even more crucial and exert a stronger effect in these patients. Our use of a high-income setting odds ratio may therefore not only be appropriate but a conservative estimate of the true value for LMICs. Please see the "discussion" section for details. Our model may be updated with an LMIC-specific odds ratio when available. |
| Costs of care in pneumonia can be used to approximate costs of care in sepsis. | Detailed costing data on sepsis-specific healthcare utilization are not available in Uganda, to our knowledge. Pneumonia features similar healthcare needs as sepsis (antibiotic, oxygen, fluids) and may progress to sepsis. Therefore, we will use previously published Ugandan costing data for pneumonia to approximate the care costs of sepsis [23], and update our model with all-cause sepsis costs when available. |
| The cost of death is taken to be zero in our model. (Note that care costs leading up to death are not taken to be zero and are counted in our costing data.) | In costing studies [25], OOP costing for death, such as funeral expenses, was foregone due to emotional trauma experienced by caregivers when asked such questions. Therefore, this data is not available. |
| Estimates of life expectancy from the World Health Organization, used to calculate YLL averted, representatively captures individuals with a history of childhood infection or sepsis. | Sepsis survivors are likely well-represented in census-based estimations of population life expectancy given a high childhood incidence of sepsis in Uganda. In addition, patients presenting to JRRH with infection vary widely in their initial severity of disease, and given JRRH's wide catchment area, likely reflects the spectrum of childhood infection severity. Therefore, use of census-based estimates of life expectancy to calculate YLL averted for the JRRH sample seems appropriate. |

This 7-day horizon is also congruent with the recommended duration of antibiotic administration for a typical episode of sepsis [4]. Model structure and key assumptions are presented in Fig 1 and Table 1, respectively.

## Model inputs

**Probabilities.** Probabilities for admission, receiving sepsis bundle within one hour, post-discharge healthcare utilization, and post-discharge mortality will be obtained from the Smart Triage clinical trial.

**Program costs.** Equipment costs of Smart Triage will be recorded as prices in Ugandan Shilling (UGX) if a component was available in the country; otherwise, the imported price by the research team will be used. Wages will be recorded in local currency. Since the results of this cost-effectiveness analysis are meant to inform implementation of Smart Triage as a new standard of care, we will model for continuation of Smart Triage in our base case analysis. Specifically, one-time costs will be annuitized with a 3% discount rate as per international

modelling guidelines [27], assuming a five-year lifetime for equipment, while annual costs will be distributed evenly among each patient.

As Smart Triage is being implemented at JRRH, a public center, program costs borne by the research team will be regarded as costs from the government perspective. We will assume a +/- 25% variation in a deterministic distribution to account for uncertainty in a probabilistic sensitivity analysis.

**Healthcare utilization costs.**   Government and OOP costs of care from inpatient and outpatient cases are available from a Ugandan costing study for pneumonia in children under five years, spearheaded by the Decade of Vaccine Economics (DOVE) project at Johns Hopkins University [23]. As costs of care for all-cause sepsis are unavailable and pneumonia has similar treatment needs to sepsis, such as oxygen and antibiotics, we will assume that DOVE's costs approximate that of all-cause sepsis. The DOVE costing study surveyed 16 public, 15 private for-profit, and 17 non-profit non-government facilities from four geographically diverse districts in Uganda (Gulu, Jinja, Mbarara, and Wakiso) using a top-down approach. This multi-district data provides an estimation of the national average costs of care in the public, for-profit, and non-profit sectors for inpatients and outpatients in Uganda, which we will link to care-seeking probabilities obtained from the Smart Triage trial. Per diem costs of inpatient care will be inferred in each sector by dividing average inpatient costs by the median length of stay. To maximize the external validity of our estimate for wages lost due to illness, we will use the national average daily wage in Uganda from the International Labor Organization [28] multiplied by the average days of missed work by caregivers in our trial.

**Adjusting costs for inflation.**   Cost data from secondary sources will be adjusted to current-year costs through the Ugandan gross domestic product (GDP) deflator from the World Bank. Program costs will be adjusted through the Ugandan GDP deflator if accrued locally, or through the GDP deflator of the country of purchase if imported, as per World Health Organization (WHO) guidelines for the economic evaluation of interventions [29].

**Clinical effect–years of life lost.**   As the Smart Triage trial is not powered to detect a difference in mortality, the difference in the proportion of children who receive a sepsis bundle within one hour will be linked to mortality via existing literature describing the impact of timely sepsis bundle on mortality [24]. We will then calculate years of life lost (YLL) from the modelled mortality rate using life expectancy data from WHO [30]. Due to the lack of disability weights and longitudinal disability data in LMIC sepsis survivors, we are unable to model years lived with disability for calculation of disability-adjusted life years (DALYs) averted, and therefore will restrict our analysis to YLLs averted.

## Model and parameter uncertainty

**One-way sensitivity analyses.**   We will explore the assumption of the continuation of Smart Triage beyond trial completion via a one-way sensitivity analysis where one-time costs are incurred within the study duration without annuitization. One-way sensitivity analyses using discount rates of 0%, 1% and 5% will also be performed as per international modelling guidelines [27].

In addition, the patient load of the JRRH outpatient department was significantly decreased during the first two months of the Smart Triage study due to the coronavirus (COVID) pandemic. Local collaborators noted that travel restrictions resulted in caregiver hesitancy to seek care for their children and estimated a more than 50% decrease in number of presenting patients per day. As Smart Triage's primary outcome, the proportion of children receiving sepsis bundle within one hour, is a process outcome that would be dependent on patient load, we will perform a sensitivity analysis to investigate any bias introduced by a reduced patient load.

Specifically, we will perform this sensitivity analysis by excluding the initial two months of patient recruitment along with any subsequent timeframe where patient load is judged by local clinicians to decrease by more than 50%.

**Probabilistic sensitivity analysis.** A probabilistic sensitivity analysis will be conducted using a Monte Carlo simulation with 10,000 iterations. Values from odds ratios and patient data will be drawn from a log-normal distribution and bootstrapping, respectively. Program costs will be drawn from a deterministic distribution with limits of +/-25%. Healthcare utilization costs will be drawn from mean and standard deviations via the method of moments, assuming a gamma distribution. If only a point estimate is available from the literature, the standard deviation will be inferred from a conservatively estimated coefficient of variation in consultation with expert opinion, followed by the method of moments as described. All statistical analyses and modelling will be performed using R [31].

## Results to be reported

The proportion of children who receive a sepsis bundle of care within one hour will be reported for both the pre-intervention and post-intervention phases. Likewise, all other probabilities and costs used as model inputs will be reported, including program costs. The results of the Monte Carlo simulation will be presented on a cost-effectiveness plane, in order to illustrate the percentage of simulations that result in added effectiveness for an added cost. We will summarize these results on a cost-effectiveness acceptability curve, which will illustrate the confidence of Smart Triage being cost-effective at different thresholds of willingness to pay per YLL-averted. All results will be reported in multiples of Uganda's GDP and in United States dollars (USD) for ease of interpretability by local policymakers and the international research community alike.

## Discussion

### Strengths & implications of study

The Smart Triage program is a pioneering triage infrastructure to expedite sepsis treatment using a mobile-based algorithm derived from existing guidelines combined with organizational assistance through an automated patient tracking system and an electronic clinician dashboard. The results of our proposed economic evaluation can further encourage follow-up trials and governmental scale-up efforts, especially when combined with subsequent budget impact analyses to model a district-level or national-level scale-up in Uganda [32]. To this end, we have designed Smart Triage to be of low technical maintenance after initial set up and have heavily involved local stakeholders during program implementation to ensure sustainability after study termination. This model of sustainability has been successfully used in similar projects by our group [33]. In addition, our implementation of Smart Triage as a package allows the program's various components to work in tandem to tackle multiple coexisting barriers to care. Although this implementation strategy prevents the isolated cost-effectiveness analysis of each package component, we are conceptually unlikely to exert a clinically significant effect by tackling one barrier to care but not others. Finally, our unpublished study data thus far indicates a high follow-up rate of 98.3%, minimizing the impact of incomplete outcome data in our analysis. As we relied exclusively on telephone follow-up, we optimized follow-up through rigorous checks for phone number accuracy, repeat calls to unreached participants through a standardized process, and use of shared phone number, such as that of a neighbor, should a personal number not be available. Our final follow-up rate will be published upon completion of the trial.

To our knowledge, our analysis is the first proposed study to employ a decision-analytic model in evaluating the cost-effectiveness of a sepsis program focused on triage in an LMIC setting. This technique allows for a more comprehensive evaluation of cost-effectiveness compared to previous economic analyses of LMIC triage programs [13,20], and allows for comparison of cost-effectiveness to analyses of sepsis triage programs in high income settings [18,19]. Specifically, our model allows linkage of receiving sepsis bundle within one hour to mortality based on published data, thereby increasing the clinical interpretability of our results. Additionally, a decision-analytic model allows us to incorporate post-discharge follow-up data, such as healthcare utilization and post-discharge mortality, rather than artificially restricting our analyses to the inpatient timeframe. Lastly, the use of a probabilistic sensitivity analysis incorporates uncertainty into our analysis following international modelling guidelines [27].

To enhance external validity, JRRH was selected for our study site due to the similarity of its triage infrastructure and existing use of ETAT to many Ugandan centers of comparable catchment size [34]. Therefore, despite the single-center nature of our analysis, our estimates of clinical efficacy may be suggestive of that in a typical Ugandan regional referral center, a hypothesis awaiting confirmation through a multicenter follow-up trial. Additionally, our use of a costing dataset sampled from various geographically diverse regions across Uganda also increases external validity outside of JRRH [23]. Although this costing data arises from pneumonia cases, the similar treatment requirements to sepsis make this data the best option in the absence of data for all-cause sepsis. While healthcare utilization and associated costs of pneumonia may not be identical to sepsis, the two disease processes have significant overlap as pneumonia may progress to sepsis if sub-optimally treated, particularly in resource-constrained settings. Therefore, we chose to use secondary data for pneumonia care costs in their entirety, rather than an item-by-item selection of costs. To this end, our model could be updated with LMIC-specific all-cause sepsis costing data when available. Finally, to further optimize external validity, we used the national Ugandan average in daily wages, rather than the local average, to calculate the cost of lost wages from days of missed work [35]. Through these uses of multiregional and national linkage data to maximize external validity, our analysis is well-situated to inform further multicenter research from a clinical and economic standpoint.

As Smart Triage is being implemented during the COVID-19 pandemic, we have also taken steps to maximize our analysis's external validity to the post-pandemic setting. To this end, we will perform a sensitivity analysis where time periods with decreases in estimated patient load greater than 50% will be excluded. These time periods may bias our results by artificially increasing proportion of children receiving sepsis bundle within one hour due to a lower triage burden not seen outside of a pandemic setting, or may conversely result in more extensive delays due to lower staffing ratios from staff redeployment. According to local staff, these decreases in patient load only occurred within the first two months of patient recruitment as of this protocol's writing and may have resulted from caregivers' hesitancy to seek care due to travel restrictions early in the pandemic. We were also informed by local administrative and clinical staff that while JRRH has put in place infectious disease precautions in response to the pandemic, the triage process remains unchanged from the pre-pandemic process. Therefore, changes observed in the clinical trial may be more easily attributable to Smart Triage rather than another change to the triage system made in response to the pandemic.

## Limitations of study

Our linkage of receiving sepsis bundle within one hour to mortality provides a more clinically relevant endpoint to our analysis by calculating cost per YLL averted. However, we use data

from high-income settings for this linkage, as LMIC-specific data is not yet available due to the recency of the one-hour recommendation for defining timely treatment [4]. Despite this apparent limitation, we postulate that this substitution is appropriate, as the relationship between timely sepsis bundle and mortality in a high-income setting is likely a conservative estimate of that in the LMIC setting for several reasons. Firstly, children are more likely to present later in their disease course due to pre-hospital barriers to care such as delayed care-giver recognition of illness or inefficient transportation [36]. Secondly, children in LMICs have a lower physiological reserve from a higher comorbidity burden and malnutrition rate. Lastly, these children may present in greater need of resuscitation than a high-income setting, due to the preponderance of dehydration from diarrheal diseases [3] aggregated by decreased access to clean drinking water. For reasons such as these, early sepsis bundle may conceptually be more crucial and exert an even greater mortality benefit on children in LMICs than high-income settings. Therefore, as a corollary, not only is our substitution with high-income setting linkage data appropriate, but a future repeated analysis with LMIC-specific data may reveal even greater cost-effectiveness than our proposed analysis. On the other hand, if we had foregone the linkage of a timely sepsis bundle to mortality due to the current unavailability of LMIC-specific linkage data, our analysis would generate only the cost per child receiving timely bundle, as opposed to cost per YLL-averted, with a significantly reduced clinical interpretability.

Another area where LMIC-specific evidence is lacking is long-term quality-of-life data and corresponding disability weights in sepsis survivors. Therefore, while cost per DALY averted is a widely used metric to calculate cost-effectiveness in the LMIC setting, we are unable to estimate years lived with disability for calculating DALYs. Extrapolating quality-of-life evidence from high-income settings into the LMIC setting would be inappropriate given vast differences in social support systems. We also lacked the resources to carry out our own LMIC-based valuation study where different health states are methodically assigned disability weights. When such evidence with appropriate valuation data becomes available in the LMIC setting, long-term outcomes may be incorporated into our economic evaluation through a Markov model. An estimate of the cost per DALY averted from such a model would enhance comparability with estimates of the cost-effectiveness of other common health systems interventions in LMICs.

Lastly, while our cost-effectiveness analysis will be an important first step in shaping local policy towards scaling up of Smart Triage should cost effectiveness be demonstrated, our analysis does not take into account the impact of scale up on issues such as reallocation of resources, impact on healthcare budget given sepsis' burden of disease [32], and multicenter scale-up costs [37]. To this end, future directions of study may include a budget impact analysis factoring in scale-up costs, the number of centers for scale up, and the large burden of disease of sepsis in these centers.

## Conclusions

Conducting a cost effectiveness analysis of a program like Smart Triage is an important aspect of any parent study, especially in resource constrained settings. To complement the Smart Triage interventional trial, aimed at providing expedited sepsis treatment in children, we propose methods for a cost-effectiveness analysis of the platform. Our analysis, together with subsequent analyses modelling budget impact and scale up, will inform future modifications to the Smart Triage platform, as well as motivate scale-up to the district and national levels. This protocol may also serve as a template for the economic evaluation of similar technologies in LMICs.

## Supporting information

**S1 Table. Key assumptions of the decision analytic model.**
(DOCX)

## Acknowledgments

The authors would like to thank collaborators from Walimu and Jinja Regional Referral Hospital.

## Author Contributions

**Conceptualization:** Edmond C. K. Li, Abner Tagoola, Clare Komugisha, Annette Mary Nabweteme, J. Mark Ansermino, Craig Mitton, Niranjan Kissoon, Asif R. Khowaja.

**Data curation:** Edmond C. K. Li, Sela Grays, Annette Mary Nabweteme.

**Investigation:** Abner Tagoola, Clare Komugisha, Annette Mary Nabweteme, J. Mark Ansermino.

**Methodology:** Edmond C. K. Li, Sela Grays, Abner Tagoola, Craig Mitton, Niranjan Kissoon, Asif R. Khowaja.

**Project administration:** Edmond C. K. Li, Sela Grays, Abner Tagoola, Clare Komugisha.

**Resources:** Edmond C. K. Li.

**Supervision:** Edmond C. K. Li, J. Mark Ansermino, Craig Mitton, Niranjan Kissoon, Asif R. Khowaja.

**Visualization:** Edmond C. K. Li.

**Writing – original draft:** Edmond C. K. Li, Sela Grays.

**Writing – review & editing:** Edmond C. K. Li, Sela Grays, Abner Tagoola, Clare Komugisha, J. Mark Ansermino, Craig Mitton, Niranjan Kissoon, Asif R. Khowaja.

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
