## [Decision Letter · Decision Letter 0]

2 Sep 2021

PONE-D-21-21174

Cost-effectiveness analysis protocol of the Smart Triage program: a point-of-care digital triage platform for pediatric sepsis in Eastern Uganda

PLOS ONE

Dear Dr. Grays,

Thank you for submitting your manuscript to PLOS ONE. After careful consideration, we feel that it has merit but does not fully meet PLOS ONE’s publication criteria as it currently stands. Therefore, we invite you to submit a revised version of the manuscript that addresses the points raised during the review process.

We look forward to receiving your revised manuscript.

Kind regards,

Tai-Heng Chen, M.D.

Academic Editor

PLOS ONE

Reviewers' comments:

Reviewer's Responses to Questions

**Comments to the Author**

1. Does the manuscript provide a valid rationale for the proposed study, with clearly identified and justified research questions?

Reviewer #1: Yes

Reviewer #2: Yes

Reviewer #3: Yes

2. Is the protocol technically sound and planned in a manner that will lead to a meaningful outcome and allow testing the stated hypotheses?

Reviewer #1: Yes

Reviewer #2: Yes

Reviewer #3: Yes

3. Is the methodology feasible and described in sufficient detail to allow the work to be replicable?

Reviewer #1: Yes

Reviewer #2: Yes

Reviewer #3: Yes

4. Have the authors described where all data underlying the findings will be made available when the study is complete?

Reviewer #1: Yes

Reviewer #2: Yes

Reviewer #3: Yes

5. Is the manuscript presented in an intelligible fashion and written in standard English?

Reviewer #1: Yes

Reviewer #2: Yes

Reviewer #3: Yes

6. Review Comments to the Author

You may also provide optional suggestions and comments to authors that they might find helpful in planning their study.

Reviewer #1: Thanks for the opportunity to review this manuscript entitled "Cost-effectiveness analysis protocol of the Smart Triage program: a point-of-care digital triage platform for pediatric sepsis in Eastern Uganda". This is a protocol for an important research question. Overall, the manuscript is well written with enough details in different sections. Tables are informative. Below are comments/concerns for the authors to consider.

- Include a results section with expected outcomes and results to be reported

- Include a conclusions section

- Discuss alternative strategies to address some potential limitations

- Discussion, the authors should expand and elaborate more on how their findings support or contrast available literature and provide suggestions for future research directions that would address existing knowledge gaps.

Reviewer #2: Manuscript title: Cost-effectiveness analysis protocol of the Smart Triage program: a point-of-care digital triage platform for pediatric sepsis in Eastern Uganda

The manuscript reports a study protocol. Sepsis is an important cause of mortality in children and early treatment results in better outcomes. ETAT and other guidelines when properly implemented show good predictive power for mortality and improved clinical outcomes.

The SMART Triage program tackles guideline complexity to predict, at admission, those children most likely to deteriorate. The SMART triage program consists of a mobile app linked to a dashboard and a locally hosted Bluetooth treatment tracking mechanism to improve patient organization.

The study is part of a larger study examining pediatric sepsis treatment and is done in Jinga Teaching and Referral Hospital in Uganda. Figure 1 is a useful aid in understanding where the data analysis decision points will be made.

The decision analytic model which includes a societal perspective is a valuable inclusion as out of pocket costs calculations recognize that patients’ bear a sizable portion of health care costs.

Previously published secondary data will be used to link healthcare utilization with costs and intermediate outcomes with mortality. This data is in the Harvard Dataverse repository. It requires permissions for downloading and as such, it was not available to the reviewer in a timely fashion. That said, the variables for calculation of costs seems reasonable based on the brief description available.

Monte Carlo simulations seem very appropriate for this type of analysis-though this type of analysis is outside my specific areas of expertise.

Comments:

This protocol is clearly written. Strengths and limitations are well stated. A major strength is the linking of the sepsis bundle within one hour to longitudinal clinical data and including the 7-day post discharge outpatient time frame. Another strength is the inclusion of cost data from out of pocket expenditures.

1. Please consider a wording change-(patient care organization may be better wording than patient organization)

2. The asserted high follow up rate in the pilot data is a very reassuring, but also quite uncommon (98.5%). Please describe what methods are used to ensure so little loss to follow up.

3. YLL averted is measured by a comparison of data from Jinga hospital where SMART Triage program in utilized (intervention) + one Kenyan hospital and data from one hospitals in Kenya (control).

Have the authors considered using a within Jinga hospital pre intervention time frame as a control for that comparison as well? YLL averted is the critical variable on which all other cost data is framed. Variation in implementation of the ETAT protocol between countries may be greater than hoped and could influence assessment of YLL averted.

Recommendation: Accept with minor revision

Reviewer #3: Thanks for the opportunity to review this interesting protocol. The protocol is well written and detailed, and I make a few suggestions that may make the work easier to follow.

Materials and Methods

1. I suggest you summarize the predictive model the Smart Triage is based on – unless its proprietary. It would be helpful for readers. If proprietary, please say so.

2. Some abbreviations are still not written in full at first use e.g. OOP

3. Description of parent trial – provide more detail on the trial. Are patients randomized? How? Are all patients at the outpatient department screened and enrolled or is there a sampling framework? Does the trial collect data on hard outcomes such as mortality, even if it is not powered to assess them?

4. It seems that the methods make a fundamental assumption – that more children receiving the sepsis bundle is good, even if it is not indicated. Please clarify.

5. Pneumonia patients often require imaging. Will the cost data be item by item, allowing the exclusion of items that may not be relevant? Or are we assuming that it the costs where pneumonia and sepsis differ may not meaningfully matter since pneumonia often leads to sepsis too.

6. Are the cost data from public or private settings? How much variation is factored in?

7. The models assume that the life expectancy of sepsis survivors is similar to the life expectancy of the general population. Without data to back it up, this assumption seems fraught.

7. PLOS authors have the option to publish the peer review history of their article (what does this mean?). If published, this will include your full peer review and any attached files.

Reviewer #1: No

Reviewer #2: No

Reviewer #3: No

---

## [Author Response · Author response to Decision Letter 0]

14 Oct 2021

Responses to reviewer and editor comments have been written in a response letter attached to this submission.

---

## [Decision Letter · Decision Letter 1]

2 Nov 2021

Cost-effectiveness analysis protocol of the Smart Triage program: A point-of-care digital triage platform for pediatric sepsis in Eastern Uganda

PONE-D-21-21174R1

Dear Dr. Grays,

We’re pleased to inform you that your manuscript has been judged scientifically suitable for publication and will be formally accepted for publication once it meets all outstanding technical requirements.

Kind regards,

Tai-Heng Chen, M.D.

Academic Editor

PLOS ONE

Reviewers' comments:

Reviewer's Responses to Questions

**Comments to the Author**

1. Does the manuscript provide a valid rationale for the proposed study, with clearly identified and justified research questions?

Reviewer #1: Yes

Reviewer #3: Yes

2. Is the protocol technically sound and planned in a manner that will lead to a meaningful outcome and allow testing the stated hypotheses?

Reviewer #1: Yes

Reviewer #3: Yes

3. Is the methodology feasible and described in sufficient detail to allow the work to be replicable?

Reviewer #1: Yes

Reviewer #3: Yes

4. Have the authors described where all data underlying the findings will be made available when the study is complete?

Reviewer #1: Yes

Reviewer #3: Yes

5. Is the manuscript presented in an intelligible fashion and written in standard English?

Reviewer #1: Yes

Reviewer #3: Yes

6. Review Comments to the Author

You may also provide optional suggestions and comments to authors that they might find helpful in planning their study.

Reviewer #1: The authors addressed all my earlier concerns. I have no further or additional comments. The authors a great job improving the manuscript.

Reviewer #3: Thanks for addressing the comments from previous review. Looking forward to seeing the article in the world.

7. PLOS authors have the option to publish the peer review history of their article (what does this mean?). If published, this will include your full peer review and any attached files.

Reviewer #1: No

Reviewer #3: No

---

## [Editor Report · Acceptance letter]

5 Nov 2021

PONE-D-21-21174R1 

Cost-effectiveness analysis protocol of the Smart Triage program:
 A point-of-care digital triage platform for pediatric sepsis in Eastern Uganda 

Dear Dr. Grays:

I'm pleased to inform you that your manuscript has been deemed suitable for publication in PLOS ONE. Congratulations! Your manuscript is now with our production department. 

Kind regards, 

on behalf of

Dr. Tai-Heng Chen 

Academic Editor

PLOS ONE